# Stretch-activated ion channels identified in the touch-sensitive structures of carnivorous Droseraceae plants

Carl Procko[1†], Swetha Murthy[2,3†‡], William T Keenan[2,3†],
Seyed Ali Reza Mousavi[2,3], Tsegaye Dabi[1], Adam Coombs[2,3], Erik Procko[4],
Lisa Baird[5], Ardem Patapoutian[2,3*], Joanne Chory[1*]

[1]Plant Biology Laboratory, Salk Institute for Biological Studies, La Jolla, United States; [2]Department of Neuroscience, Dorris Neuroscience Center, Scripps Research, San Diego, United States; [3]Howard Hughes Medical Institute, Chevy Chase, United States; [4]Department of Biochemistry, University of Illinois at Urbana-Champaign, Urbana, United States; [5]Department of Biology, University of San Diego, San Diego, United States

*For correspondence:
ardem@scripps.edu (AP);
chory@salk.edu (JC)

[†]These authors contributed equally to this work

Present address: [‡]Vollum Institute, Oregon Health & Science University, Portland, United States

Competing interests: The authors declare that no competing interests exist.

**Abstract** In response to touch, some carnivorous plants such as the Venus flytrap have evolved spectacular movements to capture animals for nutrient acquisition. However, the molecules that confer this sensitivity remain unknown. We used comparative transcriptomics to show that expression of three genes encoding homologs of the MscS-Like (MSL) and OSCA/TMEM63 family of mechanosensitive ion channels are localized to touch-sensitive trigger hairs of Venus flytrap. We focus here on the candidate with the most enriched expression in trigger hairs, the MSL homolog FLYCATCHER1 (FLYC1). We show that *FLYC1* transcripts are localized to mechanosensory cells within the trigger hair, transfecting *FLYC1* induces chloride-permeable stretch-activated currents in naïve cells, and transcripts coding for *FLYC1* homologs are expressed in touch-sensing cells of Cape sundew, a related carnivorous plant of the Droseraceae family. Our data suggest that the mechanism of prey recognition in carnivorous Droseraceae evolved by co-opting ancestral mechanosensitive ion channels to sense touch.

## Introduction

How organisms evolve new forms and functions is a question of major interest. For example, whereas most plants respond slowly to mechanical stimuli—such as touch, gravity, or changes in turgor pressure—by altering growth, some plants have additionally evolved rapid touch-induced movements. These movements can be used to deter herbivory, as is seen with the sensitive plant (*Mimosa pudica*), or to physically capture and digest mobile animals, as occurs in carnivorous plants (*Chehab et al., 2008*). The best known carnivorous plant, *Dionaea muscipula* (Venus flytrap) of the Droseraceae family, is one such plant that has evolved a remarkable and rapid touch response to facilitate prey capture (*Video 1*). How this predator evolved its complex leaf form and function for this purpose has long puzzled scientists; indeed, Charles Darwin proclaimed the species "one of the most wonderful [plants] in the world" (*Darwin, 1875*).

Several groups have described the events that evoke trap closure in Venus flytrap in detail. The plant leaf consists of an open bilobed trap, to which animal prey are attracted by volatile secretions (*Kreuzwieser et al., 2015*; *Lloyd, 1942*). There, the animal comes into contact with one of three to four mechanosensory trigger hairs on each lobe. Bending of any one trigger hair generates an action potential in sensory cells at the base of the trigger hair that propagates through the trap (*Benolken and Jacobson, 1970*; *Burdon-Sanderson, 1873*) and is correlated with an increase in

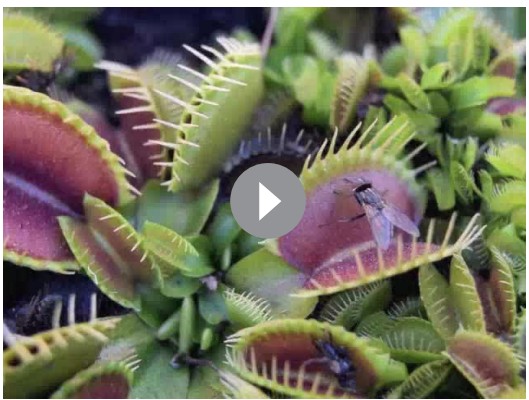

**Video 1.** Touch response of the Venus flytrap. Real-time recording of trap closure in response to insect (house fly) touch.

https://elifesciences.org/articles/64250#video1

cytosolic calcium in leaf trap cells (*Suda et al., 2020*). Two action potentials (or two touches) in a short period are required to likely further increase cytosolic calcium levels above a necessary threshold to initiate rapid closure of the trap, which ensnares the prey (*Brown and Sharp, 1910*; *Suda et al., 2020*). Trap closure occurs in as little as 100 ms, representing one of the fastest movements in the plant kingdom (*Forterre et al., 2005*). These electromotive properties were first described in the 1870s (*Burdon-Sanderson, 1873*); however, almost a century and a half later, the molecular underpinnings of these events remain elusive.

Early recordings from Venus flytrap trigger hairs suggest that, based on the time scale of action potential generation, mechanosensitive ion channels may play a role in transducing force into electrical signals (*Benolken and Jacobson, 1970*; *Jacobson, 1965*). To identify possible ion channels required for the touch response in Venus flytrap, we sought to find genes that are preferentially expressed in trigger hairs.

## Results

The Venus flytrap genome is very large (*Palfalvi et al., 2020*)—approximately 20× the size of that of the commonly studied model plant *Arabidopsis thaliana* (*Figure 1—source data 1*)—and lacks available inbred strains. Therefore, we established a clonal propagation system for the collection of genetically identical material (*Figure 1A,B*, and *Figure 1—figure supplement 1*). Using these clones, we generated a de novo transcriptome representing genes expressed in trap tissue from Illumina-based short sequencing reads. The transcriptome of our clonal strain consists of almost 28,000 unique transcripts with open reading frames coding for predicted proteins of at least 100 amino acids. This is larger than the approximately 21,000 genes previously predicted by genome sequencing (*Palfalvi et al., 2020*), reflecting the presence of multiple isoforms in our transcriptome, possible heterozygosity and strain differences, and/or differences in genome/transcriptome completeness, indicated by BUSCO analysis (see Materials and Methods). When we compared the transcriptome of trigger hairs with that of only the leaf traps, 495 protein-coding genes were differentially enriched by greater than twofold in the trigger hair, whereas 1844 were similarly enriched in the trap (*Supplementary file 1A and 1B*). Based on homology to Arabidopsis, many genes preferentially expressed in leaf traps were associated with photosynthetic function, whereas those more highly expressed in the trigger hairs included transcription factors and genes that may affect cellular and organ structure (*Supplementary file 1C and 1D*).

To find potential mechanosensitive ion channels, we screened our trigger hair-enriched transcriptome for transcripts coding for likely multi-pass transmembrane proteins. Of 45 such transcripts, three coded for possible candidates based on homology to Arabidopsis proteins. Two of these shared homology to MSL family proteins (transcript IDs comp20014_c0_seq1 and comp28902_c0_seq1), and one to the OSCA family (comp16046_c0_seq1) (*Haswell and Meyerowitz, 2006*; *Murthy et al., 2018*). We call these genes *FLYCATCHER1* and *FLYCATCHER2* (*FLYC1* and *FLYC2*) and *DmOSCA*, respectively. We did not observe enriched expression of homologs to the mechanically activated PIEZO channels. Strikingly, one of these MSL-related transcripts, *FLYC1*, was expressed 85-fold higher in trigger hairs than in trap tissue (*Figure 1C,D*), and was the second highest enriched gene (a putative terpene synthase was 176-fold enriched; see Materials and Methods). By contrast, the other two putative ion channels were less than sevenfold differentially expressed.

Using a similar approach, *Iosip et al., 2020* recently described 495 genes as trigger hair-specific. These genes were enriched in the trigger hair relative to six other tissue types: root, flower, petiole, trap, rim, and gland tissues. Remarkably, this approach identified *FLYC1/DmMSL10* (transcript

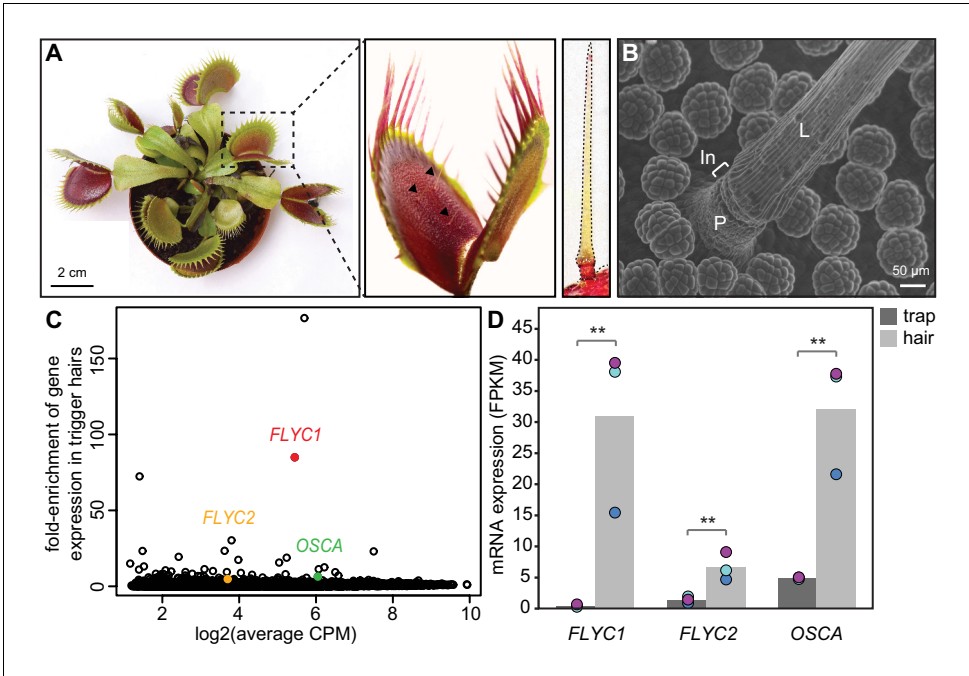

**Figure 1.** Identification of putative mechanosensory channels in the Venus flytrap trigger hair. (**A**) Representative image of a soil-grown Venus flytrap clone (left), Venus flytrap leaf (center), and single trigger hair (right). Black arrowheads in the center picture indicate trigger hairs on leaf. (**B**) Scanning electron micrograph of a trigger hair. Cells of the lever (L), indentation zone (In) and podium (P) are indicated. Also seen are the digestive glands on the floor of the lobe. (**C**) Fold enrichment of protein-coding genes of >100 amino acids in length (black circles) in the trigger hair relative to the trap. *FLYC1*, *FLYC2*, and *OSCA* are shown in red, orange, and green, respectively. CPM, counts per million of mapped sequencing reads. (**D**) Average Fragments Per Kilobase of transcript per Million mapped reads (FPKM) for *FLYC1*, *FLYC2*, and *OSCA* in traps and trigger hairs. Dots of the same color indicate paired biological replicates. **FDR < 0.005.

The online version of this article includes the following source data and figure supplement(s) for figure 1:

**Source data 1.** Size estimates of two Arabidopsis (Col-0) samples compared to our Venus flytrap strain (CP01).
**Source data 2.** Summary of sequencing reads used to build the de novo transcriptome (NCBI Transcriptome Shotgun Assembly Sequence Database accession # GHJF00000000).
**Figure supplement 1.** Venus flytrap clonal propagation system.
**Figure supplement 2.** Gene structure of Venus flytrap *FLYC1*.
**Figure supplement 3.** Gene sequence of *FLYC1*.

---

*Dm_00009130-RA*) as the most trigger hair-specific of all genes. *FLYC2* (transcript *Dm_00001246-RA*) and *DmOSCA* (3' end of transcript *Dm_00001755-RA*) were not identified as trigger hair-specific using this analysis; however, both transcripts were enriched over twofold in trigger hairs when compared to other leaf tissues (trap, petiole, rim, and glands). Expression of these genes in root and/or flower tissue might suggest that they have additional functions in other organs (*Iosip et al., 2020*). Together, these findings are consistent with our own, and support the possibility that all three genes might play a role in trigger hair touch sensation.

To validate our transcriptome, we verified the *FLYC1* transcript sequence from cDNA. In addition, we used Sanger sequencing of PCR products generated from genomic DNA template to assemble the complete gene sequence. Thirty-two single nucleotide polymorphisms (SNPs) were detected between the two *FLYC1* alleles of our clonal strain, of which only two were found in the coding region and were both silent (*Figure 1—figure supplement 2*). To further assess the natural variation that may exist, we sequenced the *FLYC1* gene from a morphologically dissimilar cultivar, and found additional SNPs and short sequence variations. Of these, eight SNPs were located in exons, two of which caused amino acid changes (*Figure 1—figure supplements 2* and *3*). No additional SNPs were found in a coding sequence reported by others (*Iosip et al., 2020*; *Palfalvi et al., 2020*). The

scarcity of SNPs causing amino acid changes when compared to the abundance of SNPs in introns is likely indicative of selection for the protein product.

*FLYC1* and *FLYC2* code for predicted proteins of 752 and 897 amino acids, respectively, with homology to Arabidopsis MSL10 and MSL5 (*Figure 2A*). Ten MSL (MscS-like) proteins have been identified in Arabidopsis based on their similarity to the bacterial <u>m</u>echano<u>s</u>ensitive <u>c</u>hannel of <u>s</u>mall conductance, MscS (*Haswell and Meyerowitz, 2006*). In *Escherichia coli*, MscS opens to allow ion release upon osmotic down-shock and cell swelling, thereby preventing cell rupture (*Booth and Blount, 2012*; *Levina et al., 1999*). In Arabidopsis, MSL8 functions in pollen rehydration, whereas the roles of the remaining MSLs in mechanosensory physiological processes is unclear (*Hamilton et al., 2015*). Notably, MSL10 forms a functional mechanosensitive ion channel with slight preference for chloride when heterologously expressed in *Xenopus laevis* oocytes (*Maksaev and Haswell, 2012*). Furthermore, along with other members of the MSL family, MSL10 accounts for stretch-activated currents recorded from root protoplasts (*Haswell et al., 2008*). Whereas FLYC1 and FLYC2 are 38.8% identical (*Figure 2—figure supplement 1*), they share 47.5 and 35.5% identity

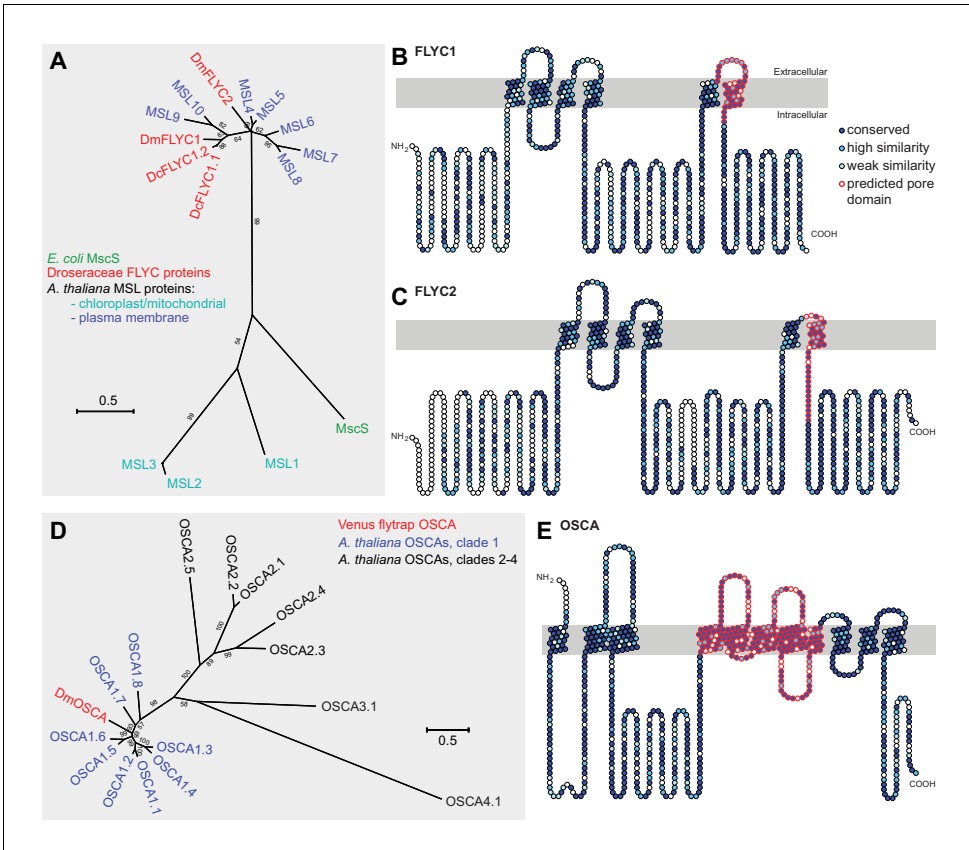

**Figure 2.** Molecular phylogenetic relationship of FLYCATCHER and OSCA proteins. (**A**) Phylogenetic analysis by maximum likelihood method to show the relationship between the conserved MscS domain (*Haswell and Meyerowitz, 2006*) of *Escherichia coli* MscS protein, *Arabidopsis thaliana* MSL proteins (MSL1–MSL10), and *Dionaea muscipula*/Venus flytrap (DmFLYC1 and DmFLYC2) and *Drosera capensis*/Cape sundew (DcFLYC1.1 and DcFLYC1.2) FLYCATCHER proteins. (**B, C**) Predicted topology for DmFLYC1 (**B**) and DmFLYC2 (**C**) proteins. (**D**) Phylogenetic analysis by maximum likelihood method showing the relationship between *A. thaliana* OSCA family proteins and Venus flytrap DmOSCA. (**E**) Predicted topology for DmOSCA protein. In (**A**) and (**D**), bootstrap values >50 are shown; scale, substitutions per site. In (**B**), (**C**), and (**E**), amino acid residues that are conserved with Arabidopsis MSL10, MSL5, and OSCA1.5, respectively, are indicated in dark blue circles, whereas residues similar in identity are indicated in lighter blue circles. The predicted pore domain for each protein is indicated in red circles.

The online version of this article includes the following figure supplement(s) for figure 2:

**Figure supplement 1.** Sequence alignment of DmFLYC1 and DmFLYC2 proteins.

with MSL10, respectively, with most variation in the cytoplasmic N-terminus. Compared to MscS, FLYC1 and FLYC2 have three additional predicted transmembrane helices, six in total, with the C-terminus of the proteins sharing highest homology (*Figure 2B,C*).

*DmOSCA* encodes a predicted 754-amino acid protein with highest homology to Arabidopsis OSCA1.5 (64% identity), which belongs to a 15-member family of OSCA proteins in Arabidopsis (*Figure 2D,E*). Although initial identification characterized the OSCA family as hyperosmolarity-activated calcium channels, it has been demonstrated that several members of the OSCA family are mechanosensitive ion channels that are non-selective for cations with some chloride permeability (*Murthy et al., 2018*; *Yuan et al., 2014*). Fly and mammalian orthologues, *Tmem63*, also encode mechanosensitive ion channels, suggesting that the molecular function of the OSCA genes is conserved. Furthermore, purification and reconstitution of AtOSCA1.2 in proteoliposomes induce stretch-activated currents, indicating that these proteins are inherently mechanosensitive (*Murthy et al., 2018*). Notably, mutant OSCA1.1 plants have stunted leaf and root growth when exposed to hyperosmotic stress (*Yuan et al., 2014*), possibly as a consequence of impaired mechanotransduction in response to changes in cell size. This suggests an ancestral role for these channels as osmosensors, similar to the MSL family.

We sought to find which cells in the trigger hair express *FLYC1*, *FLYC2*, and *DmOSCA*. The trigger hair can be divided into two main regions: a cutinized lever and a podium on which the lever sits (*Figure 1B*). An indentation zone at the top of the podium separates the two regions (*Figure 3A,C*), and is where most flexure of the trigger hair occurs (*Lloyd, 1942*). Electrophysiological recordings have demonstrated that mechanical stimulation of the trigger hair generates action potentials from a single layer of sensory cells at this indentation zone, and not from other cells of the podium or lever (*Benolken and Jacobson, 1970*). Remarkably, by fluorescent in situ hybridization, we detected *FLYC1* transcript specifically in indentation zone sensory cells (*Figure 3B,D*). No transcript was observed in cells of the lever or lower podium (*Figure 3B,E*). In contrast, *EF1α* transcript, a housekeeping gene, was detected throughout the trigger hair and trap (*Figure 3—figure supplement 1*). *FLYC1*-sense probes produced no signal above background (*Figure 3B*). Using similar in situ hybridization methods, we were unable to detect *FLYC2* transcripts, whereas *DmOSCA* was found at high levels within trigger hair sensory cells, but also low levels in other cell types (*Figure 3—figure supplement 2*). These results are consistent with our RNA-seq findings from whole trigger hairs, where *FLYC2* and *DmOSCA* were less enriched over background trap tissue compared to *FLYC1*, and in the case of *FLYC2* only weakly expressed (*Figure 1D*). Regardless, the expression profile of *FLYC1* and *DmOSCA* is consistent with them being at the site of touch-induced initiation of action potentials.

To test whether *FLYC1, FLYC2, and DmOSCA* were indeed mechanically activated ion channels and could confer mechanosensitivity to naïve cells, we expressed human codon-optimized sequences of these genes in mechanically insensitive HEK-P1KO cells (*Dubin et al., 2017*). Robust stretch-activated currents were recorded from *FLYC1*-expressing cells when negative pressure was applied to the recording pipette in the cell-attached patch configuration (*Figure 4*, A), but not from *FLYC2*-, or *DmOSCA*-expressing cells (*Figure 4—figure supplement 1*). Notably, overexpression of human codon-optimized *MSL10* and of *OSCA1.5* subcloned from *A. thaliana* also failed to produce stretch-activated currents in our system (*Figure 4—figure supplement 1*). However, previous studies have shown that expression of *MSL10* in oocytes does elicit stretch-activated currents, suggesting that trafficking of these plant proteins is largely compromised in mammalian cells (*Maksaev and Haswell, 2012*). Given the high transcript enrichment, localization in sensory cells, and mechanosensitivity, we focused on FLYC1 as a likely functional mechanosensor in Venus flytrap sensory hairs.

Further characterization of the *FLYC1* channel in cell-attached patch clamp mode indicated that the pressure required for half-maximal activation ($P_{50}$) of FLYC1 was 77.3 ± 4.0 mmHg (N = 9) (*Figure 4B* and *Figure 4—figure supplement 2A*). The channel has a conductance of 164 ± 9 pS (N = 6) in physiological recording solution (*Figure 4C*), and upon removal of the stretch stimulus, the currents decay with a time constant of 167 ± 34 ms (N = 5). In land plants and green algae, chloride-permeable channel opening is associated with membrane depolarization, due to the efflux of chloride ions down their electrochemical gradient (*Beilby, 2007*; *Hedrich, 2012*). Therefore, we tested whether FLYC1 is permeable to chloride by recording stretch-activated FLYC1 currents from inside-out excised patches in symmetrical versus asymmetrical NaCl solution. In symmetrical 150 mM NaCl, FLYC1 stretch-activated single-channel currents reversed at +2.9 ± 1.0 mV (N = 8), whereas in

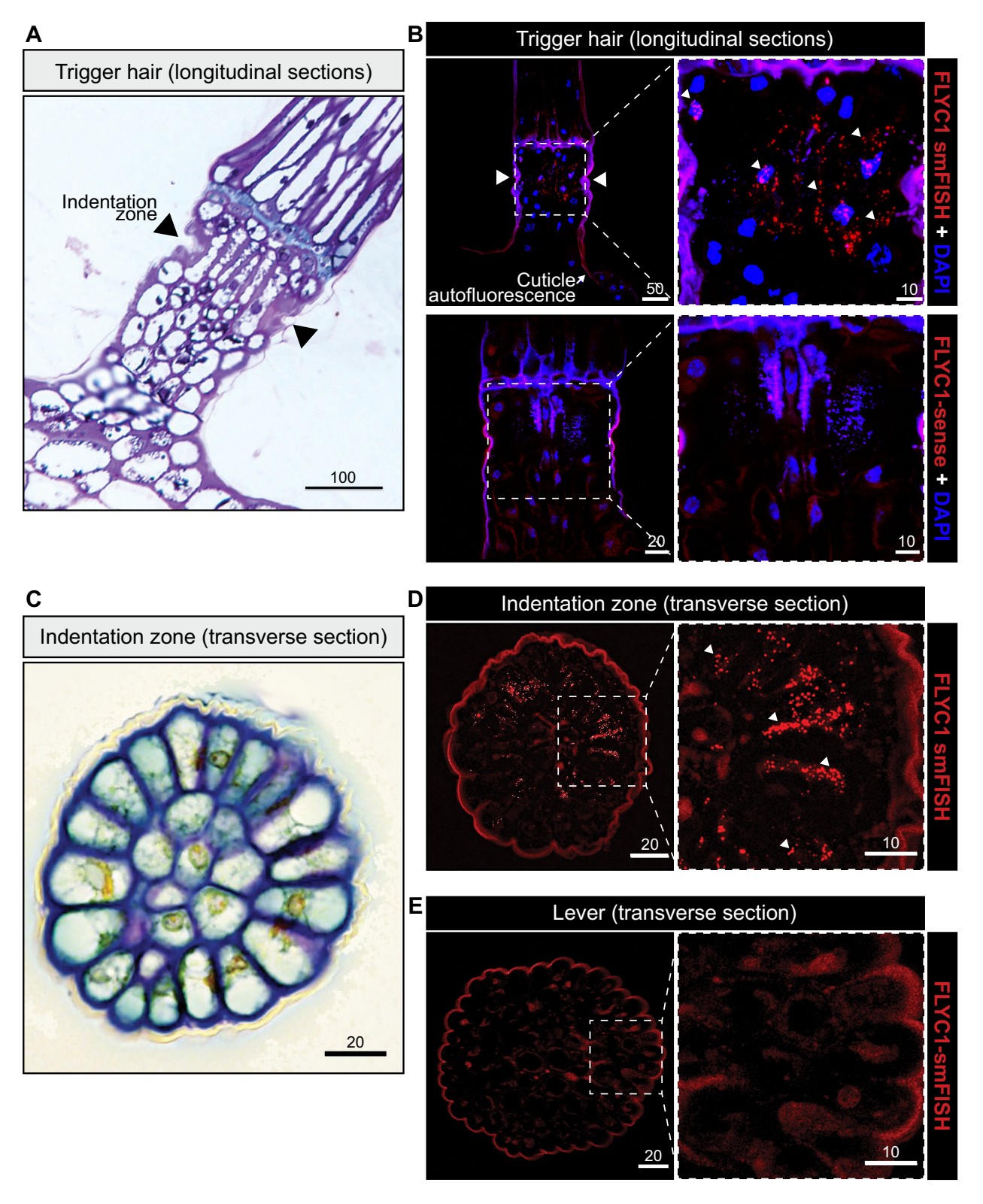

**Figure 3.** *FLYC1* mRNA localization in Venus flytrap trigger hairs. (**A**) Toluidine blue-stained longitudinal section through the base of a trigger hair. Elongated sensory cells are visible at the indentation zone (arrowheads). (**B**) Max projection through a longitudinal section after fluorescent in situ hybridization. (Top) *FLYC1* transcript (red) and DAPI (blue) at low (left) and high (right) magnification of the indentation zone. Note localization of the *FLYC1* transcript puncta in sensory cells. (Bottom) No signal was observed when using *FLYC1*-sense probes. High background fluorescence is observed

*Figure 3 continued on next page*

*Figure 3 continued*

in all channels. (C) Toluidine blue-stained transverse section through the indentation zone. The cells forming the outer ring are presumed to be the mechanosensors (**Benolken and Jacobson, 1970**). (D) Max projection through a transverse section after fluorescent in situ hybridization. *FLYC1* transcript (red) at low (left) and high (right) magnification. Note localization of the *FLYC1* transcript puncta predominantly in the sensory cells of the indentation zone. (E) Max projection through a transverse section of the trigger hair lever. No transcript was observed. Scale bars, μm.

The online version of this article includes the following figure supplement(s) for figure 3:

**Figure supplement 1.** Autofluorescence and *EF1α* transcript expression in Venus flytrap.

**Figure supplement 2.** *FLYC2* and *DmOSCA* transcript expression in trigger hair.

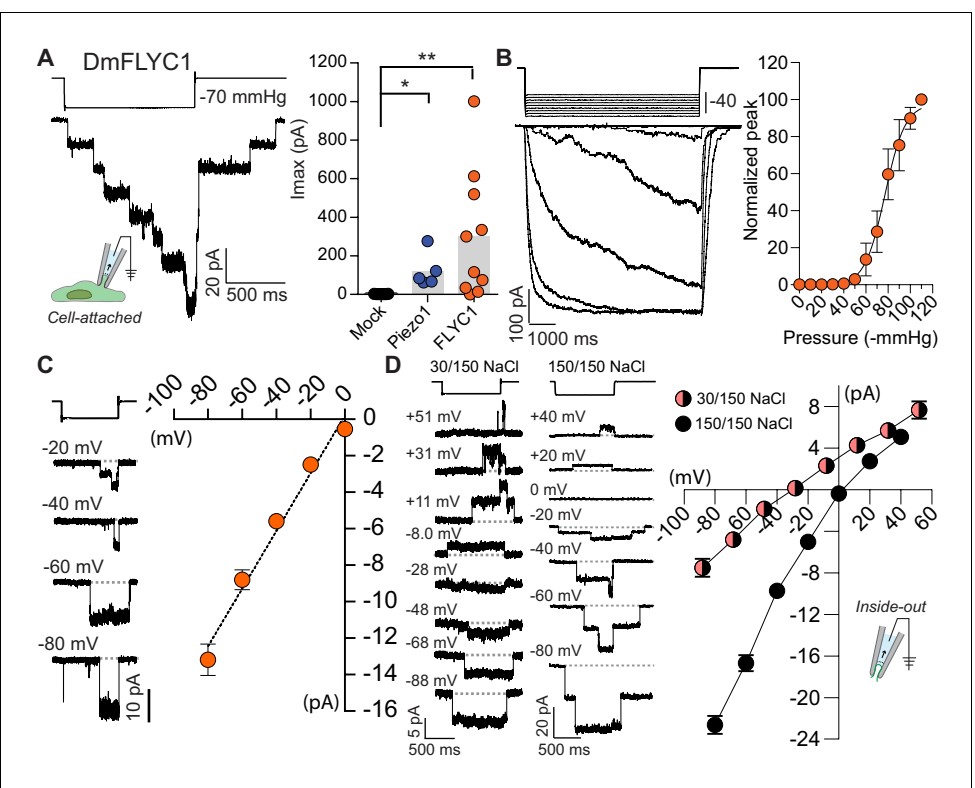

**Figure 4.** *FLYC1* induces stretch-activated currents. (**A**) Left, representative trace of stretch-activated current recorded from *FLYC1*-expressing HEK-P1KO cells in the cell-attached patch clamp configuration at −80 mV membrane potential in response to −70 mmHg pipette pressure. Stimulus trace illustrated above the current trace. Right, quantification of maximal current response from cells transfected with mock (N = 7), mouse *PIEZO1* (N = 5), or *FLYC1* plasmid (N = 10). p=0.0251 (mock vs. *PIEZO1*); p=0.0070 (mock vs. *FLYC1*); Dunn's multiple comparison test. (**B**) Left, currents in response to graded negative pressure steps from 50 to 90 mmHg (Δ5 mmHg) at −60 mV membrane potential. Right, average pressure–response curve normalized to peak current across cells (pressure–response curve for absolute peak values is plotted in **Figure 4—figure supplement 2**). A fit with Boltzmann equation revealed $P_{50}$ value of 77 mmHg (N = 9). (**C**) Left, representative single-channel traces in response to stretch at the indicated membrane potential. Right, average I–V relationship of stretch-activated single-channel currents from *FLYC1*-transfected cells (N = 6). (**D**) Left, representative stretch-activated single-channel currents recorded from excised inside-out patch configuration in asymmetrical or symmetrical NaCl solution at the indicated membrane potential. Right, average I–V of stretch-activated single-channel currents in asymmetrical NaCl solution (N = 7, red/black circles) or symmetrical NaCl solution (N = 8, black cirlces). Data for both conditions were collected from separate patches/cells. Scatter plots in B–D are mean ± s.e.m.

The online version of this article includes the following figure supplement(s) for figure 4:

**Figure supplement 1.** *DmFLYC2* and *DmOSCA* functionality.

**Figure supplement 2.** Half-maximal pressure response and chloride permeability in *DmFLYC1*.

**Figure supplement 3.** The sequence of a putative pore-forming helix in *FLYC1* is compatible with MscS-like channel structure.

intracellular 30 mM NaCl and extracellular 150 mM NaCl asymmetrical solutions, FLYC1 currents reversed at $-30 \pm 1.5$ mV (N = 7). This reversal potential is closer to the theoretical reversal potential of $-37.8$ mV for chloride ion than of $+37.8$ mV for sodium ion, derived from the Nernst equation under our experimental conditions. Therefore, FLYC1 exhibited a preference for chloride over sodium with a $P_{Cl}/P_{Na}$ ratio of $9.8 \pm 1.8$ (N = 7), as measured using the Goldman–Hodgkin–Katz equation (*Figure 4D*). These results are further supported by the lack of inward chloride currents at positive potential when stretch-activated currents were recorded in calcium-gluconate containing extracellular solution in the cell-attached patch clamp mode (*Figure 4—figure supplement 2B*). Finally, from our excised inside-out patches in symmetrical 150 mM NaCl the single-channel conductance of FLYC1 was $276 \pm 10$ pS (N = 9), calculated as the slope of the I–V curve between $-80$ and $0$ mV (*Figure 4D*).

Bacterial MscS are bona fide mechanically activated ion channels, with in-depth structural and molecular understanding of how they are gated by membrane tension (*Booth and Blount, 2012*). Much less is known about the structure and function of plant MSLs. Although homology modeling of MSL10 with MscS has hinted at certain residues that alter channel conductance, molecular determinants of selectivity and gating in MSL10 remain largely unknown (*Maksaev and Haswell, 2013; Maksaev et al., 2018*). Intrigued by the high sequence similarity in the C-terminus of MscS, MSL10, and FLYC1 proteins (*Figure 4—figure supplement 3A*), we asked to what extent the molecular architecture of the putative pore-lining helix is shared among these channels. Homology-modeling of FLYC1 with the known MscS structure (*Bass et al., 2002*) suggested that certain features of the pore-forming transmembrane domain (TM6) are indeed conserved. These include hydrophobic residues within the part of the helix that lines the pore (TM6a), and a glycine kink at G575 followed by an amphipathic helix (TM6b) that lies parallel to the inner plasma membrane surface (*Figure 4—figure supplement 3B,C*). Based on our result that FLYC1 has a higher preference to chloride, we tested whether the positively charged lysines on either side of the putative pore (K558 and K579) may confer pore properties. We substituted lysines at position 558 and 579 with glutamate and measured $P_{Cl}/P_{Na}$. Whereas selectivity for chloride in both mutants remained unchanged, the K579E mutant exhibited smaller single-channel currents at positive membrane potentials (*Figure 4—figure supplement 3D*), suggesting that K579 is indeed in the vicinity of the pore. This analysis confirms that the predicted TM6 of FLYC1 is part of the pore-lining region of the channel, consistent with mutagenesis results in MSL10 (*Maksaev et al., 2018*). These observations are further supported by the cryo-electron microscopy structure of AtMSL1, which indicates a similar architecture of the last TM of the channel (*Deng et al., 2020*). Future structural studies on FLYC1 will better resolve how selectivity is determined in these channels.

If *FLYC1* is indeed important for touch-induced prey recognition, we reasoned that its expression and function in mechanosensory structures is likely to be conserved across carnivorous Droseraceae plants. To test this, we investigated the largest genus in the family, Drosera, which includes ~200 species of sundew (*Poppinga et al., 2013*). Sundews are characterized by touch-sensitive projections on their leaf surface called tentacles (*Figure 5A*; *Darwin, 1875*; *Lloyd, 1942*; *Poppinga et al., 2013*; *Williams, 1976*). These tentacles typically secrete a glob of sticky mucilage from their head, which acts as a trapping adhesive when contacted by insect prey. Movement by the adhered insect results in action potentials along the tentacle (*Williams and Pickard, 1972*), often accompanied by radial movement of the tentacle towards the leaf center (*Darwin, 1875*). This traps the struggling insect against even more mucilage-secreting tentacles, allowing for digestion to occur (*Figure 5B* and *Video 2*).

We identified the expression of two *FLYC1* homologs in Cape sundew (*Drosera capensis*) by quantitative reverse transcriptase (qRT)-PCR (*Butts et al., 2016*). Consistent with our findings from Venus flytrap, these two transcripts had 30- to 40-fold higher expression in tentacles compared to tentacle-less leaf tissue (*Figure 5C*). The cloned cDNAs of these two genes—which we call *DcFLYC1.1* and *DcFLYC1.2*—code for almost identical predicted protein products with 96.4% identity. They share 66.2 and 65.6% identity to Venus flytrap *FLYC1*, respectively (*Figure 2A*).

Cape sundew tentacles display variation in length depending on their position on the leaf (*Figure 5—figure supplement 1*), but share a similar mechanosensory head structure, with two outer layers of secretory cells (*Figure 5D*; *Lloyd, 1942*; *Williams and Pickard, 1972*, *Williams and Pickard, 1974*). The outermost secretory cells have previously been hypothesized to also be the site of touch sensation (*Lloyd, 1942*). Not only are these cells directly exposed to the mucilage

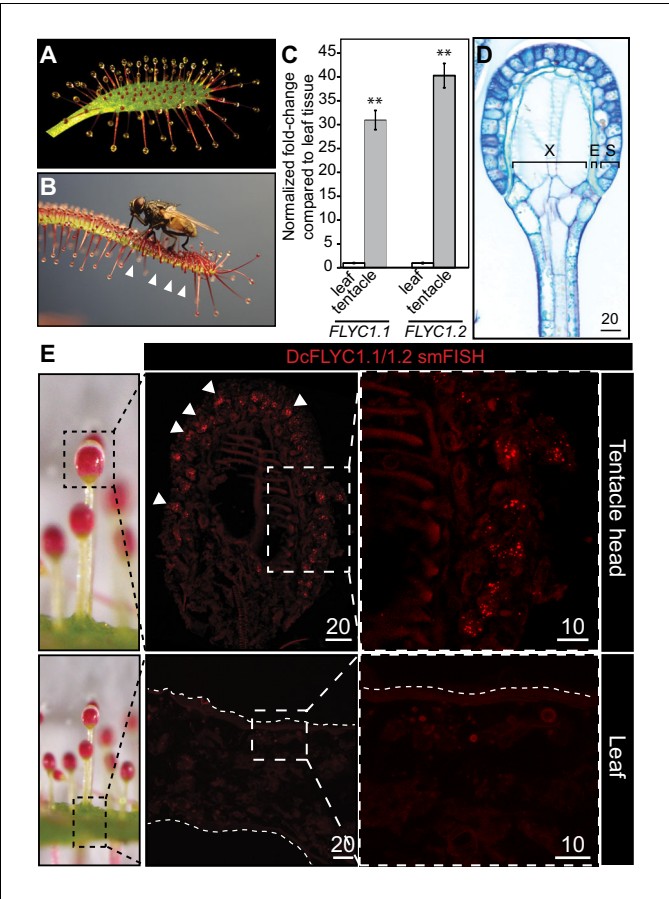

**Figure 5.** *DcFLYC1.1* and *DcFLYC1.2* localize to touch-sensitive structures of Drosera. (**A**) The Cape sundew leaf, showing tentacle projections with mucilage secretions. (**B**) Image of tentacle bending in response to insect (house fly) touch. Arrowheads mark examples of tentacles that have bent inward. (**C**) Relative expression of *DcFLYC1.1* and *DcFLYC1.2* in sundew tentacles versus tentacle-less leaves by qRT-PCR. **$p<0.005$, moderated *t*-test, results from four biological replicates of each tissue type. (**D**) Toluidine blue–stained longitudinal section through the head and upper neck of a Cape sundew tentacle. The head is composed of xylem (X), an endodermis-like layer (E), and secretory cells (S). (**E**) Max projection image, at low (left) and high magnification (right), showing collective localization of *DcFLYC1.1* and *DcFLYC1.2* mRNAs to the outer secretory cells of the tentacle head (top). Arrows indicate example secretory cells with *DcFLYC* puncta. No signal above background was observed in the leaf (bottom). Scale bar, µm.

The online version of this article includes the following figure supplement(s) for figure 5:

**Figure supplement 1.** Variation in Cape sundew tentacles.

**Figure supplement 2.** Unique cell morphology of Cape sundew sensory/excretory cells.

**Figure supplement 3.** Drosera FLYC functionality.

environment on which the insect prey adheres and pulls, but they display a unique morphology of outer cell wall buttresses and plasma membrane crenellations (*Figure 5—figure supplement 2*; *Lloyd, 1942*). In addition, cellulose fibrils extend from the outer cell wall into the cuticle in much the same way as they do in the Venus flytrap indentation zone (*Sievers, 1968*; *Williams and Pickard, 1974*). Strikingly, smFISH probes against *DcFLYC1.1* and *DcFLYC1.2* transcripts localized to the outer secretory cells, whereas no transcripts were observed in the leaf at the base of the tentacles (*Figure 5E*).

Heterologous expression of human codon-optimized *DcFLYC1.1* and *DcFLYC1.2* cDNA, independently or together in HEK-P1KO cells, did not result in stretch-activated currents (*Figure 5—figure supplement 3*). Similar to our findings with *DmFLYC2* and *DmOSCA*, we speculate that the lack of activity could be due to incorrect folding and trafficking of these proteins. Nonetheless, our

expression data is consistent with a conserved role for *FLYC1* in two divergent species in carnivorous Droseraceae plants.

## Discussion

Here, we identify members of the *FLYC* and *OSCA* families of ion channels as candidate mechano-sensors in rapid touch sensation in carnivorous plants. First, members of both families have enriched mRNA transcript expression in the sensory trigger hairs of Venus flytrap. Among these, *FLYC1* is a prime candidate as its mRNA is massively enriched in the putative mechanosensory cells that initiate transduction of the touch-induced signal. Second, *FLYC1* forms a mechanically activated ion channel with properties that would facilitate generation of action potentials in sensory cells. Third, expression of *FLYC* genes is remarkably conserved in two morphologically disparate touch-sensitive structures from different genera in the Droseraceae family. In addition to the three candidates presented here, our results do not exclude the possibility that other mechanosensitive proteins could also contribute to the events that enable touch-induced prey capture. In the future, knockout experiments will be critical to determine which candidate mechanosensor(s) identified here, if any, are absolutely necessary for functional prey detection and trap closure.

Using a mammalian cell-expression system, we detected robust macroscopic as well as single-channel stretch-activated currents in FLYC1-expressing cells and identified residues important for channel properties, which strongly suggests that FLYC1 form mechanosensitive ion channels. However, we were unable to detect currents from the other channels tested, including Arabidopsis MSL10, which has previously been shown to induce currents only in oocytes but not in mammalian cells. We hypothesize that these channels fail to fold and traffic correctly to the mammalian cell membrane, or, alternatively, that the activation threshold for these channels might be higher than that of DmFLYC1 (Venus flytrap), technically precluding us from recording reliable stretch-activated currents. Future functional studies on these genes in different non-mammalian expression systems and purification and reconstitution of proteins in lipid vesicles will conclusively determine whether these genes encode inherently mechanosensitive ion channels.

It is interesting to compare FLYC1 channel properties to its homologs, MSL10 and MscS. FLYC1 single-channel conductance is $164 \pm 9$ pS (in physiological pipette solution) in cell-attached patches and $276 \pm 10$ pS (in 150 mM NaCl) in excised inside-out patches from HEK-P1KO cells, whereas MSL10 and MscS channel conductance from oocytes is $103 \pm 3$ pS and $218 \pm 2$ pS, respectively (*Maksaev and Haswell, 2011*). FLYC1 and MSL10 stretch-activated currents have slow opening and closing kinetics with lack of inactivation in the presence of sustained pressure, unlike MscS (*Maksaev and Haswell, 2011*). The half-maximal activation ($P_{50}$ $77.3 \pm 4.0$ mmHg) for FLYC1 is two-fold lower than its bacterial orthologs MscS ($P_{50}$ $188 \pm 31$ mmHg; *Akitake et al., 2005*) and MscL ($P_{50}$ $147.3 \pm 4.3$ mmHg; *Doerner et al., 2012*). However, we observed that similar to MSL10 recordings in oocytes (*Maksaev and Haswell, 2012*), the cell membrane in our recordings often ruptured before current saturation (saturated open probability (Po) was measured in only 3/9 cells) resulting in our measurement of FLYC1 $P_{50}$ likely being an underestimation. Finally, $P_{Cl}/P_{Na}$ ratio of FLYC1 ($P_{Cl}/P_{Na} = 9.8$) is higher than ratios obtained for MSL10 ($P_{Cl}/P_{Na} = 5.9$) and MscS ($P_{Cl}/P_{K} = 1.2–3.0$) (*Cox et al., 2014*; *Maksaev and Haswell, 2012*). In-depth functional and structural characterization of these channels (FLYC1, MSL10, and MscS) under similar expression systems and recording conditions will further determine absolute differences in their properties and the underlying molecular architecture that govern these properties.

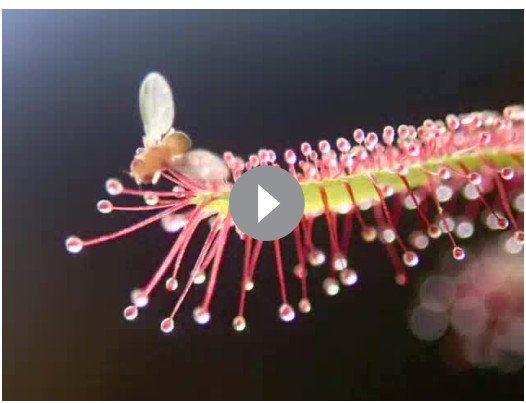

**Video 2.** Touch response of the Cape sundew plant. Time lapse recording of tentacle bending in response to insect (*Drosophila melanogaster*) touch. Insects were placed directly on the leaf tentacles and the recording started within 10 s thereafter. 1 s represents 5 min.
https://elifesciences.org/articles/64250#video2

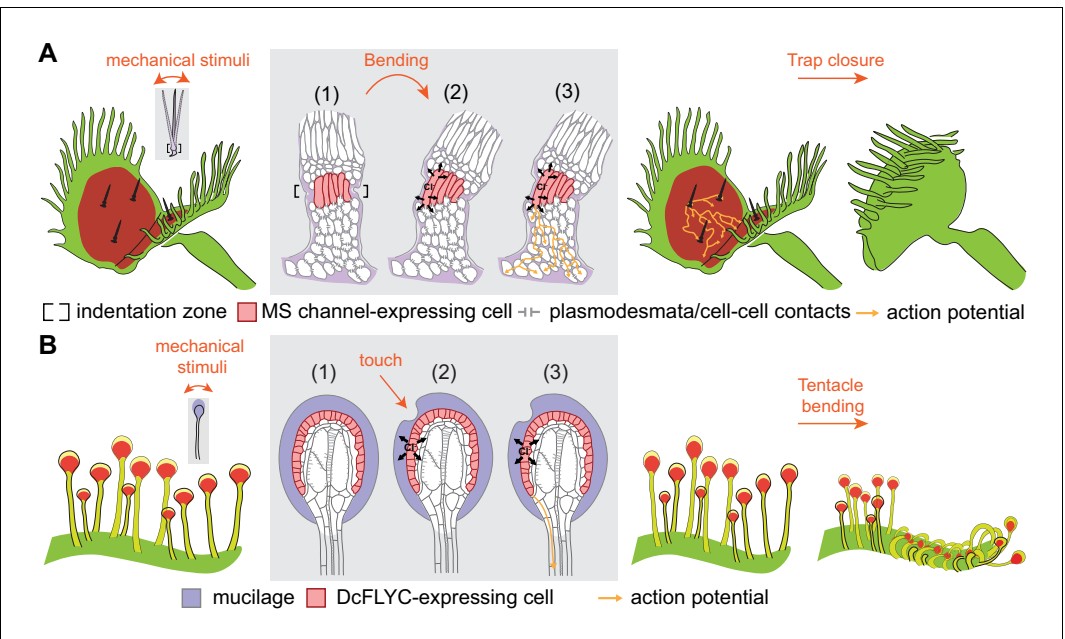

**Figure 6.** Model of touch-induced movements in carnivorous Droseraceae plants. (**A**) In Venus flytrap, mechanical stimulation of the trigger hair by a prey animal causes bending at the indentation zone sensory cells (1), leading to mechanically induced activation of mechanosensitive (MS) channels and chloride efflux (2). This triggers an action potential that propagates from the base of the sensory cells to cells of the podium via plasmodesmata (3) (***Williams and Mozingo, 1971***). Propagation of action potentials through the lobe of the leaf results in trap closure. (**B**) In sundew tentacles, the pulling of mucilage could lead to activation of DcFLYC1.1 and DcFLYC1.2 proteins in the outer cell layer of tentacle heads, triggering a propagating action potential down the tentacle stalk.

Our characterization of FLYC1 as a chloride-permeable mechanically activated ion channel suggests a possible mechanism to elicit rapid touch-induced movements in Venus flytrap. Due to the high electronegativity of sensory cells (***Benolken and Jacobson, 1970***; ***Hodick and Sievers, 1988***), combined with a concentration gradient that likely favors efflux (***Higinbotham et al., 1967***), a chloride-permeable channel like FLYC1 could contribute to membrane depolarization. This is consistent with the finding that increasing concentrations of extracellular chloride ions reduced or abolished the electrical response of Venus flytrap sensory cells to a mechanical stimulus (***Jacobson, 1974***). FLYC1-mediated depolarization, likely in combination with FLYC2 and OSCA, would then initiate action potentials that propagate into the leaf through plasmodesmata clustered on the basal side of the sensory cells (***Williams and Mozingo, 1971***), eliciting trap closure (***Figure 6A***). The role of a mechanosensitive calcium permeable ion channel, OSCA, is noteworthy here. Indeed, it has been described that trigger hair movement causes a rise in cytosolic calcium and that this signal propagates through the trap preceding its closure (***Suda et al., 2020***); however, the propagation signal most likely involves a mechanism independent of mechanotransduction. A similar chain of events would occur in the sundew tentacle: prey contact with the tentacle results in DcFLYC-induced action potentials, possibly in combination with OSCA or other mechanosensory proteins not identified here, that may propagate through plasmodesmata along the outer cell layers of the tentacle mechanosensory head and stalk, evoking tentacle bending (***Figure 6B***).

A method for transformation of Venus flytrap has been recently reported (***Suda et al., 2020***). By generating targeted knockouts using this method, future studies may be able to address which or if all of these channels are required redundantly for touch responses in the plant. Nonetheless, our findings are consistent with FLYC1 being one of the bona fide sensors of touch in Darwin's 'most wonderful' plants, and shed new light on how carnivorous plants evolved to sense touch by co-opting ancestral, osmosensing mechanosensitive channels.

## Materials and methods

### Plant materials and growth conditions

Venus flytraps were propagated in tissue culture using methods modified from those previously described (*Jang et al., 2003*). Seeds of mixed genotypes (FlytrapStore.com) were surface-sterilized for 5 min using 70% ethanol with 0.1% Triton X-100; rinsed; and then seeded onto 1/3x Murashige and Skoog (MS) salts and vitamins (Caisson Labs), 3% sucrose, and 4.3 g/L gellan gum. Following germination, plants were selected for robust growth in tissue culture as well as large size to facilitate tissue manipulations. A single strain originating from a single seed (CP01) was chosen for further propagation by continual splitting of rhizomes. After 9–12 months in culture, the largest rosettes were transferred to soil (fine-grade sphagnum peat moss) and grown under greenhouse conditions (25–30°C; ~16:8 hr light–dark cycle, with overhead artificial red light added in evening hours to bring to 16 hr light length). Soil was kept constantly moist using purified water. Plants were allowed to harden on soil for at least 2–3 months prior to experiments.

*D. capensis* var. *rubra* (Cape sundew) seeds (AK Carnivores, HI) were surface sterilized, germinated on plates, and seedlings transferred to soil and grown under greenhouse conditions.

### Imaging

For imaging of cell wall-stained tissue sections, freshly harvested plant tissue was dissected in 2% (v/v) paraformaldehyde and 1.25% (v/v) glutaraldehyde in 50 mmol/L PIPES buffer, pH 7.2 and fixed for 2 hr in the same solution. Samples were dehydrated in a graded ethanol series and embedded in JB-4 Plus Embedding Media (Electron Microscopy Sciences) according to manufacturer's instructions, with the exception that infiltration was performed at room temperature over 7 days. Some dehydrated samples were then stained with 0.1% eosin in 100% ethanol prior to embedding to help visualize the material during sectioning. Sections were cut at 4–6 µm and dried from a drop of $dH_2O$ onto Probe On Plus slides (ThermoFisher Scientific). Tissues sections were stained with 0.01% aqueous Toluidine Blue O, cover-slipped, and examined with an Olympus BX51 microscope.

Scanning electron microscopy was performed by imaging fresh plant tissue in the variable-pressure mode of a field emission-scanning electron microscope (Sigma VP; Zeiss) at 5 Pa of nitrogen with the variable-pressure secondary electron detector.

For time-lapse movies and light images of *D. capensis* during feeding, plants were fed with *Drosophila melanogaster* (Canton-S) or house flies (*Musca domestica*) and imaged using an Olympus Tough TG-5 camera. To aid in feeding, all insects were momentarily paralyzed by placing them in a tube on ice for 5–10 min immediately prior to plant feeding.

### Estimation of nuclear genome size

Nuclear genome size was estimated using flow cytometry methods similar to those reported previously (*Arumuganathan and Earle, 1991*), and were performed at Benaroya Research Institute, Seattle, WA. Nuclei from 50 mg of Arabidopsis leaf or Venus flytrap petiole tissue were suspended in 0.5 mL solution of 10 mM MgSO4, 50 mM KCl, 5 mM Hepes pH 8.0, 3 mM dithiothreitol, 0.1 mg/mL propidium iodide (PI), 1.5 mg/mL DNAse free RNAse (Roche), and 0.25% Triton X-100. Nuclei were filtered through a 30 µm nylon mesh and incubated at 37°C for 30 min. Stained nuclei were analyzed with a FACScalibur flow cytometer (Becton-Dickinson). As an internal standard, samples included nuclei from chicken red blood cells (2.5 pg/2C). For each measurement, the PI fluorescence area signals (FL2-A) from 1000 nuclei were collected and analyzed and the mean positions of the G0/G1 peaks for the sample and internal standard were determined using CellQuest software (Becton-Dickinson). Nuclear DNA size estimates are an average of four measurements.

### Tissue collection for RNA-seq and qRT-PCR

To generate RNA-seq libraries from trigger hair tissue, trigger hairs were collected over the course of a month, 3–4 times a week during the hours of 6:30–9:30 pm under artificial red light. Only traps larger than 1.5 cm in length were used. Twelve trigger hairs were dissected from the surface of two leaves at a time before snap-freezing in liquid $N_2$ (<5 min between dissection and freezing). Aliquots of trigger hairs were later pooled during RNA extraction. A sampling of traps (~20) with the trigger hairs removed was collected simultaneously as comparison tissue. The experiment was performed in

triplicate: 250 trigger hairs were collected for the first replicate and 750 trigger hairs for each of replicates two and three.

For fed versus unfed trap samples, each tissue sample included two traps (>1.5 cm in length), each fed with a single house fly for 24 hr. Fed traps were opened at the time of harvest and the fly carcass removed before snap-freezing the tissue. Samples were collected in duplicate.

Cape sundew tentacles were harvested by removing fresh leaves from plants and placing these directly into liquid $N_2$. While immersed in the liquid $N_2$, the leaves were agitated and/or scraped to break off the tentacles. The tentacles and remaining leaf material were then separated for RNA extraction. Each sample type was collected in quadruplicate.

## RNA extraction for qRT-PCR and RNA-seq

Total RNA was extracted using a modified LiCl-based method similar to that described by others (*Bemm et al., 2016*; *Jensen et al., 2015*). Briefly, frozen plant material was ground to a powder and 700 µL of RNA extraction buffer added (2% CTAB, 2% polyvinylpyrrolidone K25, 100 mM Tris HCl pH 8.0, 25 mM Na-EDTA pH 8.0, 2M NaCl, 2% v/v β-mercaptoethanol). If necessary, tissue aliquots were pooled at this stage, or later during ethanol precipitation. Samples were vortexed 2 min and then incubated 65°C for 10 min, vortexing occasionally. Debris was removed by centrifugation and 600 µL chloroform was added to the supernatant and mixed. The sample was centrifuged at 10,000 rpm for 10 min. About 1/3 vol 7.5 M LiCl was added to the aqueous phase, which was then incubated overnight at 4°C with gentle mixing. RNA was pelleted by centrifugation at 10,000 rpm for 10 min at 4°C. The RNA pellet was washed with 70% ethanol, air-dried, and re-suspended in 100 µL $H_2O$. The RNA was then re-precipitated by adding 0.1 vol 3M NaAcetate (pH 5.2), 2.5 vol 100% ethanol and an optional 1.5 µL GlycoBlue (Thermo Fisher) for low yield samples; mixing; and incubating for 1 hr at −80°C. The RNA was pelleted by centrifugation at 12,000 rpm for 20 min at 4°C. The pellet was washed with 70% ethanol, air dried, and re-suspended in 30–50 µL $H_2O$. These samples were then used directly to generate cDNA for qRT-PCR. Alternatively, samples for RNA-seq were re-suspended in 22 µL $H_2O$, and to this was added 2.5 µL 10x TURBO buffer and 1 µL TURBO DNase (Thermo Fisher). Samples were incubated at 37°C for 20–30 min. About 2.5 µL inactivation reagent was then added, and samples incubated for a further 5 min at room temperature, mixing occasionally. The resin was removed by centrifugation at 10,000 rpm for 1.5 min and transferring the supernatant to a new tube. RNA quality and yield was assessed by Agilent TapeStation before proceeding with sequencing library generation.

## RNA-Seq and Venus flytrap trap transcriptome assembly

Stranded mRNA-Seq libraries were prepared using Illumina TruSeq Stranded mRNA Library Prep Kit according to the manufacturer's instructions. Libraries were quantified, pooled, and sequenced at paired-end 125 bp reads using the Illumina HiSeq 2500 platform at the Salk NGS Core. Raw sequencing data was demultiplexed and converted into FASTQ files using CASAVA (v1.8.2). All samples were sequenced simultaneously on a single flow cell lane. The average sequencing depth was 8.8 million reads per library. Library quality was assessed using FastQC (v0.11.5) (*Andrews, 2010*), and Illumina adapter sequences and poor quality reads trimmed using Trimmomatic (0.36.0) with the suggested parameters by Trinity (*Bolger et al., 2014*).

Most aspects of our de novo transcriptome assembly and RNA-Seq analysis were performed using CyVerse cybercomputing infrastructure (*Goff et al., 2011*). So as to acquire as complete a representation of the Venus flytrap trap transcriptome as possible, we included reads collected from house fly-fed and unfed traps, in addition to our paired trigger hair and trigger hair-less trap samples to build our trap transcriptome (*Figure 1—source data 2*). We used Trinity (v2.5.1) for the de novo build (*Grabherr et al., 2011*), using default settings and an assembled contig length of greater than 300 nt. Our final transcriptome included 80,592 contigs, which were grouped into 77,539 components. In our manuscript and hereafter, each Trinity component is generally referred to as a 'gene'. The maximum contig length was 15,873 nt; average and median lengths 867 and 536 nt, respectively; and N50 1197 nt.

We used Core Eukaryotic Genes Mapping Approach (CEGMA) and Benchmarking Universal Single-Copy Orthologs (BUSCO) analyses to assess the completeness of our Venus flytrap trap transcriptome (*Parra et al., 2007*; *Simão et al., 2015*). Using CEGMA, of 248 ultra-conserved core

eukaryotic genes tested for, all were present in our transcriptome, including 246 of which the coding sequences were defined as complete by CEGMA criteria. When we searched for the presence of near-universal single copy orthologs using BUSCO (v3.0; lineage: plantae; species: Arabidopsis), 94.7% of BUSCOs were present, 92.6% of which were defined as complete. This number is greater than that reported for the previously published Venus fly trap genome sequence (*Palfalvi et al., 2020*).

To further test the quality of our transcriptome, we aligned reads for each sample back to the transcriptome using Bowtie 2 (v2.2.4) (*Langmead and Salzberg, 2012*). For each sample, 80–90% of the paired reads mapped back concordantly, while the overall alignment rate was >90%. These values were similar when the reads for each sample from a second sequencing run were independently aligned (see below).

TransDecoder (v1.0) (*Haas et al., 2013*) was used to find open reading frames (ORFs) that coded for possible proteins or incomplete protein fragments of at least 100 amino acids in length on the + strand. In total, 33,710 ORFs fulfilling these criteria were found (this number increased only a small amount, to 34,080, if the - strand was also included). Of these, 14,886 were identified as complete. Our values are not that dissimilar for a previously reported Venus flytrap transcriptome generated from multiple tissues (*Jensen et al., 2015*).

To assign homologous sequences from Arabidopsis to our protein-coding transcripts, we blasted all complete and partial polypeptides against the Arabidopsis TAIR10 proteome using default settings in Blastp (v2.2.29), with an e-value threshold <0.01 (*Camacho et al., 2009*). The top hit only was retained for downstream analysis. To predict the number of transmembrane passes per protein, we used TMHMM v. 2.0 (*Krogh et al., 2001*; *Sonnhammer et al., 1998*).

Our de novo Venus flytrap trap transcriptome is available through the National Center for Biotechnology Information (NCBI) Transcriptome Shotgun Assembly (TSA) database with accession number GHJF00000000. The uploaded transcriptome has the following modifications from that described above and used for differential gene expression analysis in the following section. The last 42 nt of contig comp11005_c0_seq1 and the first 21 nt of comp46326_c0_seq1 were removed (possible adapter sequences), and 18 contigs were flagged as possible contaminants and deleted. The uploaded transcriptome includes 80,574 contigs.

## RNA-seq differential gene expression analysis

We used methods supported by Trinity (*Haas et al., 2013*) for finding genes differentially expressed between trigger hair and trigger hair-less trap tissue samples. To increase the sequencing read count number over any given contig, we performed a second sequencing run of all samples (SRA accession numbers SRR8834210, SRR8834209, SRR8834208, SRR8834207, SRR8834212, and SRR8834211). We trimmed the concatenated reads from both sequencing runs using Trimmomatic as described above, and aligned these to our de novo Venus flytrap trap transcriptome using Bowtie (*Langmead et al., 2009*). RSEM (v1.2.12) was used to find expected gene counts (*Li and Dewey, 2011*), and edgeR to find differentially expressed genes using a paired experimental design for statistical testing (*Robinson et al., 2010*). For our analysis in edgeR (v3.12.1), we first filtered our gene list to include only those Trinity components whose transcript(s) included an ORF coding for a protein (or fragment thereof) of at least 100 amino acids in length. For components with more than one such protein assigned to them, the longest ORF was assumed to be the most relevant, and was used as the basis for assigning an Arabidopsis homolog to the gene/component. Genes that had counts per million (CPM) <2 in over half the samples were excluded from the analysis. A table of genes passing these criteria showing gene expression values as mean CPM in traps and trigger hairs, as well as differential expression in trigger hairs versus traps calculated using edgeR algorithms with false discovery rate (FDR) can be found at the NCBI Gene Expression Omnibus (GEO) database (*Edgar et al., 2002*) with GEO Series accession number GSE131340 (https://www.ncbi.nlm.nih.gov/geo/query/acc. cgi?acc=GSE131340). This repository also includes a list of all protein-coding genes and blast results against the Arabidopsis proteome. Genes that had a fold enrichment difference of twofold or more, in addition to FDR <0.05, were designated as having trigger hair- or trap-enriched expression. These are shown in *Supplementary file 1A and B*. Note that fold enrichment is calculated using edgeR algorithms, and not directly from mean CPM values. Using predictive fold changes for each of the three replicate pairs in edgeR (prior.count = 1), our three genes of interest were also all consistently enriched in the trigger hairs. Specifically, *FLYC1* was the fourth, second, and third most enriched

gene in replicate pairs 1–3, respectively, while *FLYC2* was 57th, 259th, and 57th, and *DmOSCA* 108th, 41st, and 40th most enriched. For gene ontology (GO) analysis (*Supplementary file 1C and D*), GO terms were assigned to each gene based on its homologous Arabidopsis protein (Arabidopsis TAIR10 annotation data downloaded, April 2018). GO enrichment was performed using BiNGO 3.03 (biological process terms only) with Benjamini and Hochberg corrected p value < 0.05 (*Maere et al., 2005*).

The highest differentially expressed gene in our trigger hair transcriptome coded for a protein with homology to a terpene synthase (transcript ID comp18811_c0_seq1; approximately 175-fold enrichment). It is possible that the expression of this gene is a vestigial remnant of the proposed evolutionary history of the trigger hair from an ancestral, tentacle-like secretory structure (*Williams, 1976*). However, we do not see enriched expression of other genes obviously associated with volatile production. Interestingly, when we blasted the transcript against the Arabidopsis TAIR10 genome (https://www.arabidopsis.org/Blast/) we found a reported hit against Arabidopsis MSL10, due to a conserved 17 nt stretch. Thus, a shared regulatory sequence may exist between this transcript and MSL-related genes.

## Cloning of Venus flytrap cDNAs and sequencing of *DmFLYC1* genomic DNA

cDNA from Venus flytrap whole-trap tissue was prepared from RNA using the Maxima First Strand cDNA Synthesis Kit (Thermo Fisher). *FLYC1* cDNA (Trinity ID #comp20014_c0_seq1) was then PCR amplified using primers CP1010 and CP1011, which anneal in the predicted 5' and 3' UTRs, and was ligated into pCR-Blunt II-TOPO (Invitrogen). A clone was generated that matched the predicted sequence from the de novo transcriptome, as assessed by Sanger sequencing methods. Similarly, *EF1α* (comp10702_c0_seq1) was PCR amplified from cDNA template using primers CP1144 and CP1145, and ligated into pCR-Blunt II-TOPO. We chose this EF1α transcript as a ubiquitously expressed control gene for our RNAscope experiments due to its high expression in all RNA-seq tissue samples. We sequenced the *EF1α* cDNA insert using flanking M13F and M13R primers that anneal in the pCR-Blunt II-TOPO vector, and verified that the first ~1.1 kb and last ~350 bp of the cDNA matched the prediction from our de novo transcriptome.

Venus flytrap genomic DNA from strain CP01 was extracted from fresh plant tissue using a CTAB-based method (*Murray and Thompson, 1980*). Using this as template, we PCR-amplified overlapping fragments covering the gene and sequenced these using Sanger sequencing. Overlapping and nested primer sets were: CP1010 and CP0995; CP0994 and CP1034; CP1033 and CP1035; CP1172 and CP1173; CP1176 and CP1177; and CP1174 and CP1009. The presence of two overlapping peaks on a chromatogram was used as evidence of SNPs and allelic variation. Our most 3' primer (CP1009) anneals over the stop codon and the last 31 bases of the coding sequence, and as such, we cannot rule out the existence of additional SNPs within this 31 bp stretch. To compare our sequence to that of another strain, we used similar methods using the cultivar Creeping Death (FlytrapStore.com), which has elongated petioles. At sites of short insertions and/or sequence variation, we assumed similarity to the previously sequenced alleles to construct the most probable sequences. In addition, we aligned our coding sequence against previously reported transcript *Dm_00009130-RA* (*Iosip et al., 2020*; *Palfalvi et al., 2020*; https://www.biozentrum.uni-wuerzburg.de/carnivorom/resources). The three ambiguous nucleotides in *Dm_00009130-RA* were not included in the analysis, although are sites where we see allelic variation in our own sequenced strains. For electrophysiological characterization, *FLYC1*, *FLYC2*, and *OSCA* cDNAs were gene synthesized (human codon optimized) into pIRES2-mCherry vector from Genewiz. K558E and K579E substitutions in *FLYC1* were generated using Q5 Site-Directed Mutagenesis Kit (New England BioLabs) according to the manufacturer's instruction and confirmed by full-length DNA Sanger sequencing.

## Primer sequences

| Primer | Sequence (5'→ 3') |
| --- | --- |
| CP0994 | CCAGTGTCACCTTATAGGGAAGAAGCG |

*Continued on next page*

*Continued*

| Primer | Sequence (5'→ 3') |
|--------|-------------------|
| CP0995 | CCCTCGACGTAGTTCCCCTAGC |
| CP1009 | cacatcgatTCACATATTGCGGATACTAATTTCTTGGGGC |
| CP1010 | acacccgggGCTAGCTTTCATCCACCGAATAAACACC |
| CP1011 | cacatcgatACATCATTGACCAGAAGCAAGGCACTC |
| CP1033 | TGGCATCTTCATTCCATTTGAATAGGTTCTTTG |
| CP1034 | GCATACCCGACATGGTCGACAAGC |
| CP1035 | TGACAAGTTTGTTTAACTGCTTCACTGCTG |
| CP1144 | AGGTCTTTAGATTAACTCTTCAACATGGGTAAGG |
| CP1145 | ACAAGACTTCATTTTGCACCCTTCTTTATCG |
| CP1172 | GATTGAGCAAACAAAAGGCGCATGAAG |
| CP1173 | AAATTTTGACCTACGTTGACCGTCAGC |
| CP1174 | TGATCAGGCTGCGCTTAAACATGC |
| CP1176 | ACATTCTACTTTGTTTGCAATTGTTTTCCCACTC |
| CP1177 | AAGAAACATTAAGCTGCACCTGCTCC |
| CP1208 | GAGAGGTCCACCAACCTTGACTGG |
| CP1209 | AGCAACGGTCTGACGCATGTCC |
| CP1218 | GTGCCAGTGGGAAGAGTTGAGAC |
| CP1219 | CAGAGAAAGTCTCGACAACCATGGG |
| CP1224 | CAAATCATTGAAAACGTGAAGGGAAGCACTG |
| CP1225 | CCTATGAGATACTTAGCCTGTTAGCCATGC |
| CP1233 | GCCCTTCTATAGTAGTCTCACCTCTTCG |
| CP1237 | GCCATGCGCAGCATATGTACTAGC |
| CP1240 | GCTATTTCTTATTCTCCTGAGCACAACATACTG |
| CP1242 | TGATCGCTGTGTCGTAGATGGAACAATG |
| CP1243 | CTAATGGATTGCAAACTAGGAGATGCTTAGC |

## Identification of *D. capensis* cDNA sequences and design of qRT-PCR primers

To find Drosera *FLYC1* homologs, we blasted the first exon of Venus flytrap *FLYC1* against the scaffold assemblies of the *D. capensis* genome (*Butts et al., 2016*). We reasoned that because the first exon codes for the N-terminal cytoplasmic domain—which is most divergent among MSL family members—it would therefore best differentiate among different *MSL* genes and find the best *FLYC1* homolog. We found four close homologous sequences using this method (e-value of 0.0; >60% query cover; NCBI blastn), two of which were located on scaffold LIEC01006169.1, one on LIEC01010092.1, and another on LIEC01012078.1. Position 42843–46002 of scaffold LIEC01006169.1 (reverse strand; start to stop codon) and position 23270–26408 of scaffold LIEC01012078.1 (forward strand; start to stop codon) were predicted to code for transcripts closely resembling the complete Venus flytrap *FLYC1* sequence based on inferred exon structure and conserved sequences. The intervals between the start and stop codons of the two *D. capensis* sequences shared 93.7% identity, while the 3.5 kb regions immediately upstream and the 10 kb regions immediately downstream of the start and stop codons shared only 56.61% and 46.69% identity, respectively, as determined by alignment using the default settings of Clustal Omega (*Sievers et al., 2011*). This indicates that the two sequences are likely two different genes that arose from an ancestral gene duplication event. Here, we call these genes *DcFLYC1.1* and *DcFLYC1.2*, respectively.

To determine the exact *DcFLYC1.1* and *DcFLYC1.2* sequences coded for in our *D. capensis* plants, the cDNAs were amplified using primers predicted to bind in the 5' and 3' UTRS (primers CP1240 and CP1243 and CP1224 and CP1225, respectively). Each cDNA was amplified from

template derived from two different plants, and these PCR products were independently cloned into pCR-Blunt II-TOPO vector. For *DcFLYC1.1*, 1 of 2 and 2 of 2 clones from each of the two reactions from independent templates shared the same cDNA sequence as determined by Sanger sequencing. For *DcFLYC1.2*, 3 of 7 and 4 of 8 clones from each of the two reactions shared the same cDNA sequence. These sequences have been submitted to GenBank. Other clones had additional base differences not seen in any other clone. While these might represent endogenous variation in this tetraploid species, they are more likely a result of PCR amplification errors or low-fidelity transcription and reverse transcription processes.

*DcFLYC1.1* and *DcFLYC1.2* cDNAs share a high sequence similarity. To resolve between the two by qRT-PCR, we designed primer pairs where one of the two primers annealed in a less-conserved stretch of residues in the 5′ or 3′ UTR (CP1242 and CP1237 and CP1224 and CP1233, respectively). The resolution of the two cDNAs using these primers was confirmed by directly sequencing the two PCR products by Sanger sequencing. *DcFLYC1.1* and *DcFLYC1.2* cDNAs were gene synthesized by Genewiz (human codon optimized) and subcloned into pIRES2-mCherry vector for electrophysiological characterization.

To design qRT-PCR primers for the *D. capensis* EF1α reference transcripts, we amplified a PCR product from cDNA template using primers CP1208 and CP1209. Sanger sequencing of this product suggested the presence of at least two highly similar *EF1α* transcripts, as evidenced by double peaks on the chromatogram. These may represent the products of different genes or alleles. To avoid biases against any one *EF1α* transcript, we refined our qRT-PCR primers to anneal over unambiguous bases (CP1218 and CP1219).

## Quantitative real-time polymerase chain reaction

About 500–1000 ng of RNA extracted from Cape sundew leaf and tentacle tissue were used to generate cDNA using the Maxima First Strand cDNA Synthesis Kit (Thermo Fisher) according to the manufacturer's instructions. Diluted cDNA samples were then used as template for qPCR using SYBR Green/Fluorescin dye on a CFX384 Real-Time PCR Detection System (Bio-Rad). The PCR cycling condition consisted of 45 cycles of 95°C for 10 s, 60°C for 20 s, and 72°C for 30 s. Data were analyzed with Bio-Rad CFX Manager software (version 1.6), and fold-changes in gene expression were calculated using the $\Delta\Delta Ct$ method (*Schmittgen and Livak, 2008*). Comparisons were performed in biological quadruplicate.

## Bioinformatics analyses: homology modeling, protein/nucleotide comparisons, protein topology predictions, and evolutionary history

Unless otherwise specified, all nucleotide and amino acid comparisons were made using the default settings of MUSCLE (*Edgar, 2004a*, *Edgar, 2004b*). For phylogenetic tree analysis of MscS domain and OSCA proteins shown in *Figure 2A,D*, maximum likelihood trees were built from MUSCLE alignments of MscS domains (*Haswell and Meyerowitz, 2006*) and full-length OSCA proteins using MEGA X software (LG + G model, four discrete Gamma categories) and viewed using iTOL (*Kumar et al., 2018*; *Letunic and Bork, 2019*). All positions with <95% site coverage were eliminated (partial deletion option). The trees with the highest log likelihood and bootstrap values from 1000 replications are shown.

Protein topologies of FLYC1 and FLYC2 (*Figure 2B,C*) were predicted using Protter (*Omasits et al., 2014*). DmOSCA topology (*Figure 2E*) was determined by aligning the protein against Arabidopsis OSCA1.2 and assigning TM and pore domains at the same positions identified in the OSCA1.2 protein structure (*Jojoa-Cruz et al., 2018*). To model the Venus flytrap FLYC1 TM6 putative pore domain, residues A555-P593 were threaded to residues Q92-F130 of MscS in a closed conformation (PDB 2OAU) (*Bass et al., 2002*). C7 symmetry was imposed using Rosettascripts (*Fleishman et al., 2011*), and side chain and backbone conformations were minimized using the Rosetta energy function (*Leaver-Fay et al., 2011*) with the solvation term turned off due to exposed hydrophobics in the partial structure.

## Accession numbers

RNA-Seq data, our de novo transcriptome, ORF identification, and downstream differential gene expression analysis can be found at NCBI using the accession numbers referenced above, and under

the umbrella Bioproject PRJNA530242. CDS sequences subcloned and sequenced from plant cDNA template are deposited in GenBank (*DmFLYC1*, *DcFLYC1.1*, and *DcFLYC1.2* under accessions MN096566, MN096567, and MN096568, respectively).

## Cell culture and transient transfection

PIEZO1-knockout human embryonic kidney 293T (HEK-P1KO) cells were used for all heterologous expression experiments. HEK-P1KO cells were generated using CRISPR–Cas9 nuclease genome editing technique as described previously (*Lukacs et al., 2015*), and were negative for mycoplasma contamination. Cells were grown in Dulbecco's modified eagle medium (DMEM) containing 4.5 mg/mL glucose, 10% fetal bovine serum, 50 units/mL penicillin, and 50 µg/mL streptomycin. Cells were plated onto 12-mm round glass poly-d-lysine—coated coverslips placed in 24-well plates and transfected using lipofectamine 2000 (Invitrogen) according to the manufacturer's instruction. All plasmids were transfected at a concentration of 700 ng/mL. Cells were recorded from 24 to 48 hr after transfection.

## Electrophysiology

Patch-clamp experiments in cells were performed in standard cell-attached, or excised patch (inside-out) mode using Axopatch 200B amplifier (Axon Instruments). Currents were sampled at 20 kHz and filtered at 2 kHz or 10 kHz. Leak currents before mechanical stimulations were subtracted off-line from the current traces. Voltages were not corrected for a liquid junction potential (LJP) except for ion selectivity experiments. LJP was calculated using Clampex 10.6 software. All experiments were done at room temperature. Data were analyzed in Clampex 10.6 and Graphpad Prism 7.

## Solutions

For cell-attached patch-clamp recordings, external solution used to zero the membrane potential consisted of (in mM) 140 KCl, 1 $MgCl_2$, 10 glucose, and 10 HEPES (pH 7.3 with KOH). Recording pipettes were of 1–3 MΩ resistance when filled with standard solution composed of (in mM) 130 NaCl, 5 KCl, 1 $CaCl_2$, 1 $MgCl_2$, 10 TEA-Cl, and 10 HEPES (pH 7.3 with NaOH).

Ion selectivity experiments were performed in inside-out patch configurations. $P_{Cl}/P_{Na}$ was measured in extracellular solution composed of (in mM) 150 NaCl and 10 HEPES (pH 7.3 with NaOH) and intracellular solution consisted of (in mM) 30 NaCl, 10 HEPES, and 225 sucrose (pH 7.3 with NaOH). Calcium gluconate solution for *Figure 4—figure supplement 2B* was composed of (in mM) 50 calcium gluconate, 0.5 $CaCl_2$, 10 HEPES, and 170 sucrose (pH 7.3 with NaOH).

## Mechanical stimulation

Macroscopic stretch-activated currents were recorded in the cell-attached or excised, inside-out patch clamp configuration. Membrane patches were stimulated with 1 or 3 s negative pulses through the recording electrode using Clampex controlled pressure clamp HSPC-1 device (ALA-scientific), with inter-sweep duration of 1 min. Pressure for half-maximal activation was calculated by fitting pressure response curve for individual cells, which was recorded in cell-attached configuration, with Boltzmann equation and averaging $P_{50}$ values across all cells. We note that we were successful in obtaining saturating currents at higher pressures in only three out of the nine cells. Stretch-activated single-channel currents were recorded in the cell-attached configuration. Since single-channel amplitude is independent of the pressure intensity, the most optimal pressure stimulation was used to elicit responses that allowed single-channel amplitude measurements. These stimulation values were largely dependent on the number of channels in a given patch of the recording cell. Single-channel amplitude at a given potential was measured from trace histograms of 5–10 repeated recordings. Histograms were fitted with Gaussian equations using Clampfit 10.6 software. Single-channel slope conductance for each individual cell was calculated from linear regression curve fit to single-channel I–V plots.

## Permeability ratio measurements

Reversal potential for each cell in the mentioned solution was determined by interpolation of the respective current–voltage data. Permeability ratios were calculated by using the following Goldman–Hodgkin–Katz equations:

P$_{Cl}$/P$_{Na}$ ratios:

$$E_{rev} = \frac{RT}{F} ln \frac{P_{Na}[NA]_o + P_{Cl}[Cl]_i}{P_{Na}[Na]_i + P_{Cl}[Cl]_o}$$

## In situ hybridization and imaging

Whole Venus flytrap and Cape sundew leaves were cut from the plant, freshly frozen in liquid N$_2$, and 15 µm sections collected. RNA in situ hybridization (RNAscope) was performed on sections according to manufacturer's instructions (ACDBio: #323100) using probes against *DmFLYC1* (ACD-Bio; Ref: 546471, lot: 18177B), *DmFLYC1-sense* (ACDBio; Ref: 566181-C2, lot: 18361A), *DmFLYC2* (ACDBio; Ref: 546481, lot: 18177C), *DmOSCA* (ACDBio; Ref: 571691, lot: 19032A), *DcFLYC1.1/ DcFLYC1.2* (ACDBio; Ref: 572451, lot: 19037B), and *EF1α* (ACDBio; Ref: 559911-C2, lot: 18311B). *DmFLYC1-sense* (*Figure 3B*) and *DmFLYC2* (*Figure 3—figure supplement 2*) were tested on the same section. However, *DmFLYC2* probe was independently tested in two additional experiments and no signal was observed. Slides were mounted with Vectashield + DAPI (Ref: H1200, lot: ZE0815). Stained sections were imaged with a Nikon C2 laser scanning confocal microscope and z-stacks were acquired through the entire section with a 60× objective. Displayed images are max projections of the entire z-stack. Images were processed using ImageJ (Fiji image processing package).

## Acknowledgements

We thank S Morrison, K Asahina, M Stableford, S McDowell, E Perozo, and members of the Chory and Patapoutian laboratory for assistance and helpful discussions. We acknowledge Elizabeth Haswell for facilitating a collaboration between the Chory and Patapoutian laboratories. RNA-seq was performed at the Salk NGS Core Facility (funding from NIH-NCI CCSG: P30 014195, the Chapman Foundation and the Helmsley Charitable Trust) and SEM at the Waitt Advanced Biophotonics Core (funded by the Waitt Foundation and Core Grant applications NCI CCSG CA014195 and NINDS Neuroscience Center NS072031) with assistance from L Andrade.

## Additional information

### Funding

| Funder | Grant reference number | Author |
| --- | --- | --- |
| National Institutes of Health | 1F32GM101876 | Carl Procko |
| National Institutes of Health | 1RO1HL143297 | Swetha Murthy<br>William T Keenan<br>Seyed Ali Reza Mousavi<br>Ardem Patapoutian |
| National Institutes of Health | 5R35GM122604 | Carl Procko<br>Tsegaye Dabi<br>Joanne Chory |
| Howard Hughes Medical Institute | | Carl Procko<br>Swetha Murthy<br>William T Keenan<br>Seyed Ali Reza Mousavi<br>Tsegaye Dabi<br>Ardem Patapoutian<br>Joanne Chory |
| University of San Diego | Faculty Research Grant | Lisa Baird |
| George E. Hewitt Foundation for Medical Research | | William T Keenan |

The funders had no role in study design, data collection and interpretation, or the decision to submit the work for publication.

## Author contributions
Carl Procko, Conceptualization, Formal analysis, Investigation, Visualization, Methodology, Writing - original draft, Writing - review and editing, Performed gene expression analysis, cloned and generated FLYC cDNAs; Swetha Murthy, Conceptualization, Formal analysis, Investigation, Visualization, Methodology, Writing - original draft, Writing - review and editing, Recorded and analyzed electrophysiology data; William T Keenan, Conceptualization, Formal analysis, Investigation, Visualization, Methodology, Writing - original draft, Writing - review and editing, Performed and analyzed in situ expression; Seyed Ali Reza Mousavi, Conceptualization, Formal analysis, Visualization, Methodology, Writing - review and editing, Cloned and generated FLYC cDNAs; Tsegaye Dabi, Formal analysis, Investigation, Methodology; Adam Coombs, Formal analysis, Investigation, Methodology, Cloned and generated FLYC cDNAs and FLYC1 mutants; Erik Procko, Conceptualization, Formal analysis, Investigation, Methodology, Writing - review and editing, Performed FLYC1 homology modeling; Lisa Baird, Formal analysis, Investigation, Methodology, Writing - review and editing, Performed toluidine blue staining and microscopy; Ardem Patapoutian, Joanne Chory, Conceptualization, Resources, Formal analysis, Supervision, Funding acquisition, Investigation, Visualization, Writing - original draft, Project administration, Writing - review and editing

## Author ORCIDs
Carl Procko ⬩ https://orcid.org/0000-0002-4374-4283
Swetha Murthy ⬩ https://orcid.org/0000-0001-9580-3380
William T Keenan ⬩ https://orcid.org/0000-0003-3381-744X
Erik Procko ⬩ http://orcid.org/0000-0002-0028-490X
Ardem Patapoutian ⬩ https://orcid.org/0000-0003-0726-7034
Joanne Chory ⬩ https://orcid.org/0000-0002-3664-8525

## Decision letter and Author response
Decision letter https://doi.org/10.7554/eLife.64250.sa1
Author response https://doi.org/10.7554/eLife.64250.sa2

---

# Additional files

## Supplementary files
• Supplementary file 1. Identification of differentially expressed genes in the trigger hair. (A) Putative protein-coding genes with enriched expression in Venus flytrap trigger hairs. Shown are genes (Trinity components) coding for a transcript with >2-fold enriched expression in the trigger hair relative to trap tissue, and false discovery rate (FDR) < 0.05. CPM, counts per million. TM, transmembrane. (B) Putative protein-coding genes with reduced expression in Venus flytrap trigger hairs. Similar to (A), except showing genes with >2-fold reduction in the trigger hair. (C) Gene ontology (GO) enrichment analysis of genes with higher expression in trigger hairs. Shown are GO term identifiers (GO-ID) and descriptions for all GO terms that are enriched among genes with higher expression in the trigger hair relative to trap tissue (corrected p value<0.05). (D) Gene ontology (GO) enrichment analysis of genes with lower expression in trigger hairs. Similar to (C), except showing enrichment of GO terms associated with genes that have reduced expression in the trigger hair.

• Transparent reporting form

## Data availability
Raw and processed data for the Venus flytrap transcriptome assembly and its analysis are available at the NCBI Sequencing Read Archive and Gene Expression Omnibus under accession numbers PRJNA530242 and GSE131340, and the Transcriptome Shotgun Assembly at DDBJ/ENA/GenBank under the accession GHJF00000000. The transcriptome assembly described in this paper is the first version, GHJF01000000. Cloned cDNA sequences for FLYC genes are deposited at GenBank. All other data are available in the main text or supplementary materials.

The following datasets were generated:

| Author(s) | Year | Dataset title | Dataset URL | Database and Identifier |
|---|---|---|---|---|
| Procko C, Murthy SE, Keenan WT, Mousavi S, Dabi T, Coombs A, Procko E, Baird L, Patapoutian A, Chory J | 2021 | Stretch-activated ion channels identified in the touch-sensitive structures of carnivorous Droseraceae plants | https://www.ncbi.nlm.nih.gov/geo/query/acc.cgi?acc=GSE131340 | NCBI Gene Expression Omnibus, GSE131340 |
| Procko C | 2020 | Dionaea muscipula strain: CP01 Transcriptome or Gene expression | https://www.ncbi.nlm.nih.gov/bioproject/PRJNA530242 | NCBI BioProject, PRJNA530242 |

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
