## [Decision Letter]

**Acceptance summary:**

This manuscript traces an exciting quest to identify candidate mechanosensitive ion channels enabling carnivorous plants to trap insect prey. A transcriptomics project revealed candidate sensors, and functional assays demonstrated that one of them, FLYC1, forms stretch-activated channels in a heterologous system. This work sets the stage for future discovery of the signaling events that link trigger hair deflection to trap closure and the proteins that sense trigger hair deflection.

**Decision letter after peer review:**

Thank you for submitting your article "Stretch-activated ion channels identified in the touch-sensitive structures of carnivorous Droseraceae plants" for consideration by *eLife*. Your article has been reviewed by three peer reviewers, and the evaluation has been overseen by a Reviewing Editor and Kenton Swartz as the Senior Editor. The reviewers have opted to remain anonymous.

The reviewers have discussed the reviews with one another and the Reviewing Editor has drafted this decision to help you prepare a revised submission.

Summary:

This work is a fascinating story of MSL-type channel evolution toward a true touch-sensing function in plants. Reviewers and editors alike were excited about discovering which proteins might render carnivorous plants able to detect and trap insects walking inside traps. The authors present a substantial genetic/transcriptomics groundwork that allowed them to identify candidates to be channels responsible for the mechanosensory function in trigger hairs of the Venus flytrap. One of these proteins, FLYC1 appears to be the main player, acting as a bona fide homomultimeric mechanosensitive channel. The authors also describe the aspects of the cells in which FLYC1 likely functions, and suggest that these columnar cells with radially-oriented long sides presents an interesting system for future engineering-type finite-element analysis to identify zones of maximal strain/compression where the channels might be deployed for maximal efficiency.

Although there was clearly much enthusiasm, there were also some functional studies the reviewers deemed critical to complete or improve. Also, there are field standards for considering something a bona fide mechanosensitive channel that are not all met by presented data on the three channels (see details in point 3). This and lack of loss-of-function data from Venus fly traps means that the authors can't definitively conclude that these identified channels are actually responsible for the trap closure, and this needs to be acknowledged more clearly in the text.

Essential revisions:

1) All reviewers concurred that it was essential to provide a more complete electrophysiological characterization of FLYC1. Specifically, as presented in Figure 4, the channel is on the “sluggish” side. Activation takes about 500 ms, but the rate of opening, as it should, increases with tension. The authors state: “Similar to MSL10 stretch-activated currents in oocytes, FLYC1 currents had a slow rise time for activation and did not saturate, suggesting that these channels don't inactivate.” Neither of these conclusions is supported by the presented data. Extension of pressure stimuli to 2-5 seconds should produce a plateau in elicited currents, which will enable the authors to estimate the open probability (Po) and create a saturating dose-response curve. From this curve, the authors should be able to determine the midpoint of activation and the slope. Alternatively, the application of a 5s-long linear pressure ramp from 0 to ~120 mm Hg should produce a similar activation curve within one sweep. Co-expression of *E. coli* MscL in the same cell, as was previously done by David Clapham's group (Doerner, Febvay and Clapham, 2012), would enable the authors to pinpoint the tension midpoint for FLYC1 since MscL's midpoint is known (12 mN/m). Longer stimuli and special pressure protocols (Akitake, Anishkin and Sukharev, 2005) would reveal whether FLYC1 adapts or inactivates. At least a saturating dose-response curve (Po versus pressure) should be presented.

Also as related to experiments presented in Figure 4, in the Abstract it is claimed that mechanosensitive FLYC1 channels are chloride-permeable. This claim seems to depend on the reversal potential measured in Figure 4D (excised patches studied in asymmetric NaCl concentrations) and in Figure 4—figure supplement 2 (on-cell patches recording with Ca-gluconate in the pipette). The current data are weak. It would be better to perform experiments on the same samples exposed to solutions that differ only in their Cl^-^ concentration and show that the reversal potential follows the change in the Cl^-^ gradient. In principle, this could be achieved by recording currents in an excised patch exposed to a symmetric NaCl solution and then exposed to asymmetric conditions that change only Cl^-^.

2) Given the way that the authors have placed this work in a comparative analysis, it was also agreed that the authors should make an attempt to characterize the FLYC2 and DmOSCA channels in another system. The cells do not have to be Piezo1-knockouts since the conductance and kinetic properties of the plant channels are easily discernable from those of Piezos. The reviewer suggest to abandon HEK-P1KO cells, and instead try Cos, Sf9 cells, and/or oocytes. Moreover, a recording system based on msl9-1; msl10-1 knock-out plant protoplasts has been reported (Haswell et al., 2008). Δ 5 Arabidopsis plants are available now for MSL channel expression and recording. Another option is to use this system and add some electrophysiology data for the FLYC2 and DmOSCA channels.

3) Regardless of the outcome of these experiments, the authors need to directly address the fact that they will not have tested whether or not any of the three identified channel proteins are needed for trigger hair function. They remain outstanding *candidates* for this physiological role but until knock-out or knock-down experiments are performed, the possibility that none are required for his function will remain. The force-gated ion channels field has identified criteria that must be met before a protein can considered to form a bona fide mechanosensitive channel. These include: 1) the protein expressed in the correct tissue at the right time; 2) be sufficient to form a mechanosensitive channel in heterologous systems; 3) be required for physiological responses to mechanical cues. The data in this manuscript indicated that two criteria are met for FLYC1 and that only the first criterion is met for FLYC2 and DmOSCA. Yet, despite the consistent identification of these proteins as candidate mechanosensors, an implicit assumption is woven through the manuscript suggesting to readers that these three proteins are now known to be the sensors used by trigger hairs to detect insects. The authors should either be more circumspect in their writing or provide the data. Since the latter is going to be very difficult, they should leave open the possibility that FLYC2 and DmOSCA might not form mechanosensitive channels and that perhaps only a subset of the three candidates will end up being required for physiological responses. As an example, the first paragraph in the Discussion summarizes the findings but does not explicitly tell the reader that additional work is needed to determine whether any of the candidates are required for trigger hair function. Failing to explicitly state that this remains a possibility may leave readers with the impression that the problem is solved which will complicate future efforts to directly address this question by these authors and others who may be inspired to build on this study.

---

## [Author Response]

Essential revisions:1) All reviewers concurred that it was essential to provide a more complete electrophysiological characterization of FLYC1. Specifically, as presented in Figure 4, the channel is on the “sluggish” side. Activation takes about 500 ms, but the rate of opening, as it should, increases with tension. The authors state: “Similar to MSL10 stretch-activated currents in oocytes, FLYC1 currents had a slow rise time for activation and did not saturate, suggesting that these channels don't inactivate.” Neither of these conclusions is supported by the presented data. Extension of pressure stimuli to 2-5 seconds should produce a plateau in elicited currents, which will enable the authors to estimate the open probability (Po) and create a saturating dose-response curve. From this curve, the authors should be able to determine the midpoint of activation and the slope. Alternatively, the application of a 5s-long linear pressure ramp from 0 to ~120 mm Hg should produce a similar activation curve within one sweep. Co-expression of *E. coli* MscL in the same cell, as was previously done by David Clapham's group (Doerner, Febvay and Clapham, 2012), would enable the authors to pinpoint the tension midpoint for FLYC1 since MscL's midpoint is known (12 mN/m). Longer stimuli and special pressure protocols (Akitake, Anishkin and Sukharev, 2005) would reveal whether FLYC1 adapts or inactivates. At least a saturating dose-response curve (Po versus pressure) should be presented.

We would like to thank the reviewers for these great suggestions. We agree that further characterization of FLYC1 would add substantial value to the paper. We have performed and included pressure dose-response data (see new Figure 4B) that address two major properties:

Channel kinetics: As suggested by the reviewers we recorded FLYC1 currents in response to 3s pressure stimulus, which allowed the current to reach steady state. As seen in the representative trace of Figure 4B, these currents do not inactivate. We attempted several times to record currents from a longer stimulus duration (5s or 10s) from multiple cells in order to conclusively state that the channel does not inactivate over time and potentially report statistics. We also tried the linear pressure ramp from 0 to ~120 mmHg protocol as suggested by the reviewers. Unfortunately, due to the high pressure stimulus that is required to cause maximal activation of the channel, we were unsuccessful in maintaining the integrity of the patch in most of our attempts. High pressure and longer stimulus duration resulted in frequent patch rupture. Pressures beyond 100 to 110 mmHg is lytic for HEK cells. We have consistently observed this from our many recordings of stretch-activated currents from other mechanosensitive ion channels like Piezos, Tmem63s, and Arabidopsis OSCA channels. The lack of inactivation and patch rupture was also observed by Elizabeth Haswell’s group, in their recording of MSL10 from oocytes (Maksev and Haswell, 2011). Lastly, we have removed the sentence “Similar to MSL10 stretch-activated currents in oocytes, FLYC1 currents had a slow rise time for activation and did not saturate, suggesting that these channels don't inactivate.” in accordance with our inability to make such strong claims based on our presented data. However, we do acknowledge these observations in the Discussion paragraph that compares and contrasts FLYC1 properties with MSL10 and MscS.

Pressure P**_50_**: With the new 3s protocol we were successful in measuring the P50 of the channel (77.3 ± 4.0 mmHg (N=9)). Normalized and absolute values for the pressure dose-response curves are reported in Figure 4B and Figure 4—figure supplement 2A, respectively. For reasons noted above, not all cells hit saturation/ maximal open probability at higher pressure responses, but we were able to reliably record saturating currents in 3 out of 9 cells. This is indicated in the Discussion.

We hope the addition of the dose-response curve with a longer pressure stimulus is sufficient for the initial characterization of FLYC1 and allow for more detailed characterization in the future, like co-expression with *E. coli* MscL to determine absolute tension midpoint.

Also as related to experiments presented in Figure 4, in the Abstract it is claimed that mechanosensitive FLYC1 channels are chloride-permeable. This claim seems to depend on the reversal potential measured in Figure 4D (excised patches studied in asymmetric NaCl concentrations) and in Figure 4—figure supplement 2 (on-cell patches recording with Ca-gluconate in the pipette). The current data are weak. It would be better to perform experiments on the same samples exposed to solutions that differ only in their Cl^-^ concentration and show that the reversal potential follows the change in the Cl^-^ gradient. In principle, this could be achieved by recording currents in an excised patch exposed to a symmetric NaCl solution and then exposed to asymmetric conditions that change only Cl^-^.

We thank the reviewers for the insightful suggestion. We have now added additional data to strengthen our claim that FLYC1 are chloride-permeable channels. Similar to experiments done on MSL10, we recorded I-V relationship of stretch-activated single channel currents from excised inside-out patches in symmetrical NaCl. This data is added to the graph that depicted I-V relationship in asymmetrical NaCl conditions. As demonstrated in Figure 4D, changing intracellular NaCl concentration from 30 mM to 150 mM (while maintaining the extracellular solution at 150 mM NaCl) resulted in ~3 fold increase in single channel current amplitude at negative potentials and also caused a change in the reversal potential (-30 ± 1.5 mV to +2.9 ± 1.0 mV). These results are similar to those obtained for MSL10 (Maksaev and Haswell, 2011). We think that the symmetrical NaCl control, along with measured chloride permeability ratio and the calcium-gluconate experiment, provide compelling evidence to suggest that these channels are permeable to chloride. We did attempt to switch only chloride concentration on the same excised patches but these experiments proved challenging, and given the time constraint we resorted to simply recording I-V currents in symmetrical NaCl solution. Furthermore, these experiments allowed us to measure the channel’s conductance in 150mM NaCl. These results are now described in the Results section.

2) Given the way that the authors have placed this work in a comparative analysis, it was also agreed that the authors should make an attempt to characterize the FLYC2 and DmOSCA channels in another system. The cells do not have to be Piezo1-knockouts since the conductance and kinetic properties of the plant channels are easily discernable from those of Piezos. The reviewer suggest to abandon HEK-P1KO cells, and instead try Cos, Sf9 cells, and/or oocytes. Moreover, a recording system based on msl9-1; msl10-1 knock-out plant protoplasts has been reported (Haswell et al., 2008). Δ 5 Arabidopsis plants are available now for MSL channel expression and recording. Another option is to use this system and add some electrophysiology data for the FLYC2 and DmOSCA channels.

We would like to attempt expression of FLYC2 and DmOSCA in another system, since we could not get currents in our HEK expression system even with several approaches. In our past experience, if we cannot get recordings of a clone in HEK cells, it also does not work in other adherent cell lines. Consistently, we tested FLYC1, FLYC2, and DmOSCA in HeLa cells but didn’t observe stretch-activated currents from any of the candidates. We agree, however, given the success of MSL10 recording in oocytes, this approach might yield promising results. While we plan to perform these experiments, we do not currently have this type of recording set up within our labs, and standardizing and optimizing a recording system in oocytes without face-to-face instruction from a knowledgeable colleague is not practical during the COVID-19 pandemic. Therefore, attempting these experiments will cause a significant delay. We hope that the experiments we have added to further characterize FLYC1 and compare it to homologs are sufficient to highlight our focus on the most promising candidate. We do believe additional experiments on FLYC2 and DmOSCA is important for future work.

3) Regardless of the outcome of these experiments, the authors need to directly address the fact that they will not have tested whether or not any of the three identified channel proteins are needed for trigger hair function. They remain outstanding candidates for this physiological role but until knock-out or knock-down experiments are performed, the possibility that none are required for his function will remain. The force-gated ion channels field has identified criteria that must be met before a protein can considered to form a bona fide mechanosensitive channel. These include: 1) the protein expressed in the correct tissue at the right time; 2) be sufficient to form a mechanosensitive channel in heterologous systems; 3) be required for physiological responses to mechanical cues. The data in this manuscript indicated that two criteria are met for FLYC1 and that only the first criterion is met for FLYC2 and DmOSCA. Yet, despite the consistent identification of these proteins as candidate mechanosensors, an implicit assumption is woven through the manuscript suggesting to readers that these three proteins are now known to be the sensors used by trigger hairs to detect insects. The authors should either be more circumspect in their writing or provide the data. Since the latter is going to be very difficult, they should leave open the possibility that FLYC2 and DmOSCA might not form mechanosensitive channels and that perhaps only a subset of the three candidates will end up being required for physiological responses. As an example, the first paragraph in the Discussion summarizes the findings but does not explicitly tell the reader that additional work is needed to determine whether any of the candidates are required for trigger hair function. Failing to explicitly state that this remains a possibility may leave readers with the impression that the problem is solved which will complicate future efforts to directly address this question by these authors and others who may be inspired to build on this study.

The reviewers are correct that further work will be required to conclusively determine if all, some or none of these candidate mechanosensors are required for responses to insect touch in these plants. We have altered the language of the manuscript at several places to emphasize this point.

1) We have added a discussion of a paper recently published while ours was under revision (Iosip et al., 2020) which performed a similar RNAseq experiment to our own. We end this discussion with “Together, these findings are consistent with our own, and support the possibility that all three genes might play a role in trigger hair touch sensation.”

2) We have added the following to the end of the first Discussion paragraph: “In the future, knockout experiments will be critical to determine which candidate mechanosensor(s) identified here, if any, are absolutely necessary for functional prey detection and trap closure.”

3) We have highlighted the importance of further knockout studies in a newly separated final paragraph in the Discussion. 4) As suggested by reviewer 3, we have removed “resulting in trap closure” to remove the direct causal implication.